# Slovenian Higher Education in a Post-Pandemic World: Trends and Transformations

Fayyaz Qureshi [1], Sarwar Khawaja [1], Mirjana Pejić Bach [2] and Maja Meško [3,*]

1 Oxford Business College, 65 George Street Oxford, Oxford Q1 2BQ, UK; fayyaz.qureshi@oxfordbusinesscollege.ac.uk (F.Q.); sarwar.khawaja@oxfordbusinesscollege.ac.uk (S.K.)
2 Faculty of Economics and Business, University of Zagreb, Trg John F. Kennedy 6, 10000 Zagreb, Croatia; mpejic@efzg.hr
3 Faculty of Organizational Sciences, University of Maribor, Kidričeva 55a, 4000 Kranj, Slovenia
* Correspondence: maja.mesko@um.si

**Abstract:** The COVID-19 pandemic has changed many aspects of work and daily life, with higher education being greatly affected, especially in remote teaching, work, and digital collaboration. Most of these changes are retained in the post-COVID-19 era, e.g., remote work has enabled greater access to educational opportunities and contributed to a more inclusive and diverse workforce. To investigate to what extent these changes impact higher education in the post-pandemic era, we have conducted qualitative research on a sample of 12 professors from Slovenia working in higher education, selected based on their extensive research, professional experience, and significant contributions to the field. The Delphi method was used for this study since its iterative process refines ideas in each round based on feedback from the previous one. Participants were given a five-day window to express their views and share their expertise. The responses to the open-ended questions were examined using qualitative content analysis. Research indicates that pedagogical and organisational characteristics such as the ability to adapt to changes, the capacity for resilience, and the willingness to embrace digital transformation are crucial for preserving long-term changes induced by pandemics.

**Keywords:** post-pandemic world; higher education; future trends; Delphi method

## 1. Introduction

The advent and subsequent rapid spread of the COVID-19 pandemic have precipitated profound disruptions to daily routines [1,2], exerting notable influences across various spheres of global society, including Higher Education Institutions (HEIs) [3]. This unprecedented scenario has compelled HEIs to pivot towards remote learning modalities swiftly, embrace digital technologies, and recalibrate their approaches to meet the evolving needs of students [1–3]. Consequently, a critical imperative arises to scrutinise the trends and metamorphoses unfolding within higher education as we navigate a post-pandemic era. Undoubtedly, the COVID-19 pandemic has generated a multifaceted and demanding landscape for human resource management (HRM) professionals, educators, and other personnel [4]. They have been tasked with devising innovative strategies to sustain operations and support their workforce in navigating the challenges posed by the crisis [5]. The shift from conventional office settings to remote work posed a significant hurdle [4], necessitated by the severity of the COVID-19 pandemic and subsequent lockdown measures implemented in numerous countries [5].

COVID-19 research has indicated that the abrupt transition to remote work presented numerous obstacles, including work disruptions, socio-economic hardships such as job loss, and adverse effects on mental health, such as stress, anxiety, and feelings of isolation during the pandemic [2–7]. Many workers experienced notable changes in their overall digital engagement amid the COVID-19 crisis [8,9]. Amidst global lockdowns and restrictions, organisations rapidly transitioned to remote work setups, leveraging digital tools and

technology to maintain business operations and enhance collaboration [10]. Employees grappled with many digital platforms for communication and virtual meetings, marking a paradigmatic shift in their accustomed work routines. This sudden transition demanded the rapid acquisition of novel digital competencies and the adept navigation of challenges concerning technology, connectivity issues, and the delicate balance between professional responsibilities and personal wellbeing [11].

In higher education, remote work emerged as the digital remedy to sustain the continuity of learning and teaching activities [3,11]. Nevertheless, the encountered challenges were commonplace, encompassing technological hurdles (about institutional ICT resources and capacity), individual obstacles (related to technological proficiency), and difficulties in learning and teaching (such as fostering student engagement and collaborative efforts) [6]. The pandemic accelerated the adoption of online and blended learning modalities, prompting educators to rethink traditional pedagogical approaches. Research indicates a growing emphasis on active learning strategies, flipped classrooms, and asynchronous delivery methods to enhance student engagement and flexibility. Moreover, the need for personalised and adaptive learning experiences is recognised in order to cater to diverse student needs and learning styles [3,5]. The rapid shift to online learning during the pandemic underscored the importance of digital technologies in higher education. Institutions invested heavily in educational technology infrastructure, including learning management systems, video conferencing tools, and online assessment platforms [2–4,6,12]. There was a focus on leveraging emerging technologies to enhance teaching and learning outcomes [13]. Higher education institutions faced significant financial challenges during the pandemic, including a decline in enrolment, revenue losses, and budget constraints. Many institutions implemented cost-saving measures, such as faculty layoffs, programme cuts, and deferred capital projects. There was a renewed focus on financial sustainability, diversification of revenue streams, and strategic planning to build institutional resilience and adaptability in the face of future crises. Several noteworthy aspects of the digital landscape emerged throughout and following the pandemic.

For instance, remote workers encountered challenges establishing suitable home offices, managing potential distractions, and maintaining a clear boundary between work and personal life [14]. Furthermore, communication channels such as email, instant messaging, and video conferencing tools (such as Zoom and Microsoft Teams) now serve as the primary modes of interaction among colleagues, managers, and supervisors, enabling seamless virtual collaboration [3]. Employees should familiarise themselves with collaboration tools and project management platforms to efficiently manage remote tasks and projects. The global closure of educational institutions due to COVID-19 has significantly affected approximately 1.6 billion students worldwide [15]. Moreover, approximately 80–85% of students in high-income countries have transitioned to online learning, in contrast with fewer than 50% in low-income nations [16]. The widespread adoption of online learning in higher education has underscored the necessity of addressing the challenges it presents for students and educators, alongside their corresponding capabilities. Compared to traditional face-to-face settings, students enrolled in online courses are believed to be less inclined to participate in collaborative learning activities, group discussions, and interactions with instructors [17]. An expanding body of research delves into the advantages and disadvantages of online learning [17–19].

The continuous and rapid growth of Information and Communication Technology (ICT) has profoundly influenced academic discussions, standard research practices, scholarly endeavours, and teaching methods. Consequently, the utilisation and ongoing assessment of contemporary online tools have become essential for addressing students' evolving requirements, particularly within online educational contexts [20].

While higher education institutions oversee many essential digital resources utilised by the educational community, the involvement and contributions of non-academic professionals are crucial to ensuring seamless operations within the online educational environment. This includes roles that all work to facilitate the growth of the educational community.

Typically stationed at designated work locations, these professionals operate within fixed office hours and extended shifts. However, circumstances such as the COVID-19 pandemic prohibited teachers and other staff from accessing these centres, necessitating the transition to telecommuting from home. In terms of pedagogy, educators must be equipped with the necessary technical tools to enable remote access to their regular tasks, including deploying essential systems within specified timeframes [21]. Previous research indicates that the pandemic's unique circumstances have disproportionately affected teaching and learning processes [20]. While changes to teaching methods are typically voluntary and planned [22–24], emergency shifts, such as those prompted by the COVID-19 outbreak, have been less extensively studied. Many educators express regret over the pandemic's lost time and its adverse effects on the social development of young people [25].

The rapid transition to digital and online learning during COVID-19, significantly impacting academic personnel, can be attributed to several factors. Initially, the pandemic prompted educators to adopt diverse digital platforms and technologies for teaching, including online learning environments, video conferencing, and assessment systems. Simultaneously, concerns about digital access and inclusivity emerged due to the transition to online learning, as students and staff faced challenges accessing reliable internet and suitable technology. Notably, this abrupt shift to online education and remote work has increased the academic staff's workload and stress levels. Organisations can support management by implementing information systems and offering specialised training to users with varying levels of technological proficiency [3–5,26]. Collaboration with colleagues is one of the primary mechanisms through which employees navigate challenges and enhance their learning. Cooperation and communication with peers are crucial in daily tasks, and this collaboration becomes particularly important in corporate training contexts [27]. Previous research has indicated that employees' perceptions of support from their colleagues can help them adapt to new situations and alleviate stress when confronted with challenging tasks [28].

Looking ahead, it is evident that our experiences during the pandemic will significantly shape the future of work and education. The trends towards remote work and digital collaboration are expected to persist, with institutions and businesses increasingly utilising digital tools to improve productivity and collaboration. However, addressing the challenges associated with this shift is essential, including ensuring equitable access to digital resources and maintaining a work–life balance in a remote work environment. The pandemic has accelerated the digital transformation of work and education, presented significant challenges and opened up new possibilities for how we work and learn. The structure of this paper is organised sequentially. The second section provides a comprehensive review of relevant literature and theories about remote work and digital collaboration in higher education, particularly in the context of the COVID-19 pandemic. Following this, Section 3 outlines the research methods utilised to collect and analyse data for this study, detailing the research design, data collection methods, and analytical techniques. Subsequently, Section 4 presents the study's findings, while Section 5 interprets these findings with respect to the research questions and existing literature. The final section, Section 6, summarises the study's key findings and offers recommendations for future research, discussing potential implications for policy and practice in higher education.

## 2. Theoretical Background

Amid the recent pandemic, employees within the HE sector experienced a significant transformation in their digital work landscape [3,4,6]. In response to the closure of physical campuses and the need to adhere to social distancing regulations, institutions swiftly transitioned to remote and online teaching, administrative functions, and student support services [3]. A significant challenge for academics during this time was the scattered nature of literature on remote work, particularly within the digital economy [29].

Technology and digital transformation have given rise to new workplace paradigms. However, it is widely acknowledged that the physical office space will persist, as there

are distinct advantages to having co-workers gathered in a single location [30]. Extensive research has explored the advantages of remote work, including heightened productivity, enhanced knowledge sharing, increased creativity, improved employee retention, bolstered employee wellbeing, greater flexibility, and heightened job satisfaction, benefiting both individuals and organisations alike [29,31–34]. Additionally, studies have delved into the key factors conducive to the successful implementation of remote work arrangements, shedding light on the nuanced dynamics of modern work environments.

The evolution of technology and digital transformation has ushered in novel workplace paradigms. However, the enduring presence of physical office spaces remains widely recognised due to the inherent advantages of co-located co-workers [30]. Extensive research has illuminated the manifold benefits of remote work, elucidating its positive impacts on various facets of organisational functioning. These include amplified productivity, enriched knowledge sharing, heightened creativity, enhanced employee retention, fortified employee wellbeing, augmented flexibility, and elevated job satisfaction, fostering mutual gains for individuals and organisations [29,31,33,35]. Furthermore, scholarly inquiry has delved into the critical determinants that underpin the successful implementation of remote work arrangements, offering insights into the intricate dynamics of contemporary work environments and paving the way for informed decision-making in organisational settings.

Several studies have underscored the positive impact of remote work on job satisfaction, citing factors such as enhanced scheduling flexibility, strengthened collaboration, and increased knowledge sharing [36–38]. However, individuals grappling with anxiety may experience feelings of overwhelm, distress, and anxiety attributed to various factors inherent in remote work environments. These may include the demands of application multitasking, constant connectivity, information overload, frequent system upgrades, and the perpetual need for relearning, which introduce uncertainties into their roles. Moreover, technical glitches associated with the organisational use of digital technologies and platforms can exacerbate employees' distress, commonly referred to as 'technostress' [39,40].

The Technology Acceptance Model (TAM), a theoretical framework akin to this perspective, posits that individuals' perceptions of the utility and ease of use of new technologies positively influence work productivity and efficiency, thereby contributing to job satisfaction [8,18,40]. According to TAM, employees are more likely to embrace information and communication technologies (ICT), such as digital platforms, technical services, and software, if they perceive them as straightforward and beneficial tools [7]. This conceptual framework underscores the importance of perceived usefulness and ease of use in shaping employees' attitudes towards and utilisation of digital technologies in the workplace [8,41].

Various demographic factors influence the adoption of digital skills and remote work practices. Research indicates that, while men prioritise positions offering higher salary growth, women place a greater value on job flexibility and security [42]. Additionally, age is a significant determinant of attitudes towards flexible working arrangements facilitated by technology. Younger individuals are more inclined to embrace career shifts that align with their personal lives and familial responsibilities. Their digital native upbringing predisposes them to leverage technology for collaborative work, making them more amenable to remote work setups [43]. Conversely, older employees tend to emphasise the drawbacks of remote work, such as diminished interpersonal connections with colleagues and supervisors, challenges in maintaining work–life balance, and perceived inadequacies in technological proficiency [44]. These findings underscore the importance of considering demographic differences in crafting strategies to promote digital literacy and facilitate successful remote work transitions across diverse workforce populations.

Findings from a UNISON survey [45] unveiled various reasons individuals did not favour remote work arrangements. The most commonly cited reason was isolation, with 31% of respondents indicating this as a primary concern. Following closely behind, 26% of participants expressed concerns about the impact of remote work on their work–life balance. Technological difficulties were also a significant barrier, with 18% of respondents

highlighting this challenge. Additionally, 11% of participants reported needing more support as a hindrance to their remote work experience.

Furthermore, the survey revealed nuanced attitudes towards remote work among respondents. While 15% of individuals expressed a strong aversion to remote work and preferred not to continue it in the future, 12% acknowledged managing adequately but looked forward to returning to the office environment. Another 25% of respondents admitted to struggling with remote work and preferred a hybrid model combining remote and office-based work. These insights underscore the multifaceted nature of attitudes towards remote work and highlight the importance of addressing diverse concerns to facilitate successful remote work transitions.

As a result, employees are grappling with a host of challenges stemming from the necessity of transitioning to a work-from-home (WFH) setup, which has accelerated the digitalisation of human employment at an unprecedented rate [46]. This shift necessitates that employees be proficient in working and collaborating effectively in online environments, demanding specialised skills in modern office practices and efficient online communication methods. However, it is essential to recognise that specific industries, particularly those in low-skilled service sectors, may need help adopting this new paradigm due to inherent limitations. Furthermore, the feasibility of the WFH model is contingent upon factors such as network connectivity and the compatibility of tasks with online platforms, both of which can significantly impact its success [35].

In early 2020, educational institutions worldwide transitioned rapidly to remote teaching and learning, spurred by the onset of the COVID-19 pandemic. This shift underscored the importance of technology readiness and digital skills among academic staff [45]. The advent of online learning emerged as a significant alternative to the closure of educational institutions during this period, garnering attention from governments and researchers alike. Referred to as Emergency Remote Teaching (ERT) by scholars, this approach represents a temporary adaptation in educational delivery methods necessitated by emergencies. These approaches include blended learning and hybrid courses, with the intention of returning to conventional formats once the crisis abates [47]. According to a global survey by the International Association of Universities (IAU), participants reported significant impacts of COVID-19 on teaching and learning practices, with nearly two-thirds of institutions transitioning to ERT [48].

Employee technology readiness refers to employees' inclination to embrace and adopt novel technologies to enhance personal and professional objectives [49]. The adaptability of employees to new work contexts varies and is influenced by factors such as individual personalities. Proactive employees, characterised by high engagement and a drive to acquire new knowledge and skills, can effectively navigate changes in employment structures. Their proactive approach fosters an environment conducive to accommodating new requirements [50,51]. Previous research suggests that individuals with a positive attitude towards technology are more adept at acquiring new technical skills.

As a result, technological readiness drives employees to embrace new technologies and influences their behaviour towards enhanced work performance [52–54]. Adaptability emerges as another critical factor in digital readiness. Studies have underscored that employees with adaptability skills demonstrate resilience in navigating through pressures and crises. They exhibit creative problem-solving abilities, adeptly manage volatile situations, rapidly assimilate new information, and foster a culture of adaptability. Hence, understanding how to cultivate employees' adaptive performance becomes paramount, mainly through initiatives aimed at familiarising them with new technologies pertinent to their evolving job structures amidst digitalisation [55].

We have mentioned some pressing issues in the constantly evolving higher education environment. However, innovative hybrid approaches are becoming increasingly important with respect to these concerns, offering a potential way to address these issues. An emerging approach in higher education is the adoption of agile methodologies. These methodologies, inspired by software development practices, emphasise adaptability, collaboration, and

iterative learning. Agile educators focus on student-centred experiences, flexibility, and responsiveness to changing contexts. Integrating agile principles into higher education can enhance teaching, learning, and institutional resilience. Agile approaches empower educators to create dynamic, student-focused learning environments, fostering adaptability and continuous improvement [56]. Ricardo Abad Barros-Castro et al.'s theory on Computer-Supported Collaborative Learning (CSCL) within higher education introduces a systemic framework designed to evaluate CSCL initiatives, specifically focusing on mathematical problem-solving scenarios. Barros-Castro et al.'s research delves into how technology can be leveraged to enhance collaborative learning experiences in academic settings. By examining a case study in Colombia, Barros-Castro et al. shed light on practical insights and strategies for improving collaborative learning practices through integrating technology. This framework aims to assess the effectiveness of CSCL-MPS initiatives and provide valuable lessons and recommendations for educators and institutions seeking to enhance their approaches to collaborative learning and mathematical problem-solving within higher education. Barros-Castro et al.'s work underscores the importance of harnessing technology to facilitate meaningful interactions, foster collaboration, and, ultimately, optimise the learning experience for students. By offering a structured evaluation framework and drawing on real-world examples, their theory encourages adopting innovative pedagogical practices that promote active engagement, critical thinking, and effective problem-solving skills among learners in academic contexts [57].

Drawing from the theoretical underpinnings and insights garnered from the literature, this research endeavours to delve into the forthcoming trajectories and potential of remote work and digital collaboration within the realm of higher education in the aftermath of the COVID-19 crisis. The study is geared towards unravelling how the experiences encountered during the pandemic have sculpted these trajectories and discerning their ramifications for the future landscape of higher education. Furthermore, the research aims to pinpoint strategies and exemplar practices that empower higher education institutions to navigate this evolving terrain with efficacy and agility. By illuminating these aspects, the study aspires to make meaningful contributions to the ongoing discourse surrounding the digital metamorphosis of work and education. Ultimately, the endeavour seeks to furnish valuable insights that can inform policy formulation and guide practical implementations within the higher education sector, fostering resilience, adaptability, and innovation in the face of contemporary challenges and opportunities.

Given the theoretical background and the challenges and opportunities identified in the literature, this study aims to explore the future trends and possibilities of remote work and digital collaboration in higher education in the post-COVID-19 era. Specifically, the study seeks to understand how the experiences during the pandemic have shaped these trends and what implications they might have for the future of higher education. The study also aims to identify strategies and best practices to help higher education institutions navigate this new landscape effectively and efficiently. Ultimately, the goal is to contribute to the ongoing discourse on the digital transformation of work and education and provide insights that inform policy and practice in the higher education sector.

## 3. Materials and Methods

### 3.1. Sample

The sample for this study consisted of twelve professors from Slovenia who are experts in higher education. The participants included 5 women and 7 men, all from the fields of social sciences. They were selected based on their extensive research, professional experience, and significant contributions to the field. We selected professors with a minimum academic rank of associate professor and at least 10 years of teaching experience in higher education institutions to ensure they have a deep understanding of the subject matter. The participant sample consisted of 5 full professors and 7 associate professors. Additionally, we considered professors who had actively participated in online teaching or remote work during the COVID-19 pandemic to gather first-hand experiences and insights

on the challenges and successes of this transition. Their diverse perspectives and insights provided a rich understanding of the future trends and possibilities of remote work and digital collaboration in higher education post-COVID-19. This expert panel formed the basis of our Delphi study, providing valuable input and feedback throughout the research process. Their collective expertise ensured a comprehensive exploration of the topic and contributed to the validity and reliability of the study's findings.

### 3.2. Method

The Delphi method is valuable for gathering expert opinions and achieving consensus on a specific subject. This iterative process refines ideas in each round based on feedback from the previous one. Our research aimed to investigate the future of remote work and digital collaboration in higher education in the post-COVID-19 era. Expert input can offer insightful predictions and possibilities [58].

The Delphi method is commonly used to forecast future developments. It involves statistically analysing expert opinions in a particular field. This structured scientific method has well-defined rules and procedures. Experts respond individually to pre-selected questions, and an "average answer" is computed. The approach assumes no "correct" answers but provides a probabilistic estimate of the likelihood of certain events [59].

Key characteristics of the Delphi method include participant anonymity, structured feedback for experts after they provide their opinions, and the opportunity for them to revise their previous opinions until consensus is achieved. The Delphi method typically involves two to three rounds of opinion exchange between the researcher and the experts [60]. Two rounds are deemed sufficient [61,62], as additional rounds can increase the administrative load and pressure on participants, leading to lower response rates.

Loo [63] proposed that the Delphi method has the potential to predict the future of strategic management and organisational development, along with other applications in organisational management. Okoli and Pawlowski [64] recognised the Delphi method as a widely used tool in information systems research for identifying and evaluating issues related to executive decision-making. This method allows for the collection of highly reliable data from certified experts. Hence, we opted for the Delphi method in our research, as it allowed us to systematically gather and enhance professional and expert insights within this domain.

### 3.3. Procedure

The survey was conducted in two stages. The initial stage of the Delphi study posed open-ended questions about the future trends and potential of remote work and digital collaboration in higher education after COVID-19. The first question was as follows: How have the experiences during the pandemic shaped higher education trends in the post-COVID-19 era? The second question was speculative: What organisational transformations will higher education undergo in the future (5 to 10 years) due to the COVID-19 pandemic?

Participants were given a five-day window to express their views and share their expertise. The responses to the open-ended questions were examined using qualitative content analysis. Five anticipated changes were identified following the qualitative analysis of the responses received in the first stage of the Delphi study. In the second stage of the study, a structured questionnaire was developed based on the responses from the initial stage. The questionnaire included specific questions related to the key themes and trends identified during the open-ended discussions. Participants were asked to rate the importance of various factors influencing the future of remote work and digital collaboration in higher education, as well as to provide their views on the challenges and opportunities presented by these trends. The data collected from both stages of the study were analysed using qualitative and quantitative methods to identify consensus views and emerging patterns. The findings from the study were used to inform future strategies and decision-making in higher education institutions in response to the evolving landscape of remote work and digital collaboration post-COVID-19.

## 4. Results

### 4.1. First Round

Twelve professors were invited, and they all accepted the invitation. Therefore, 12 experts consented to participate in the study, and they all completed the survey in both rounds. Following the second round of the Delphi study, a moderately high level of agreement was noted.

The questionnaire was designed with two open-ended questions to prevent potential bias in the experts' responses. The primary question focused on pinpointing the elements and traits in an application that would be used to assess the applicant's ability to repay. The questions asked in the first round of the Delphi study were as follows: (1) How have the experiences during the pandemic shaped higher education trends in the post-COVID-19 era? (2) What organisational transformations will higher education undergo in the future (5 to 10 years) due to the COVID-19 pandemic?

The participants were given a five-day window to express their views and share their expertise on these topics. The second round involved specific questions presenting the aggregated responses from the first round back to the participants. They were then asked to review these responses, provide further input, or revise their previous opinions.

#### 4.1.1. Post-Pandemic Higher Education Trends

Based on the professors' responses, the study results identified several key trends shaping higher education in the post-COVID-19 era: (1) Active Learning: Higher education is shifting towards active learning, focusing on teaching skills that will remain relevant in a rapidly changing world. (2) Hybrid Learning: The pandemic has prompted a re-evaluation of the traditional concepts of time and space in education. Hybrid learning, a blend of virtual and physical classrooms, is emerging for immersive and experiential learning. (3) Digital Technologies: The rapid evolution of digital technologies, including mobile devices, cloud computing, machine learning, artificial intelligence (AI), augmented reality (AR), and virtual reality (VR), has enabled immersive and personalised online education on a larger scale and at a more affordable cost compared to traditional methods. (4) Student-Centred Collaborative Learning: There is a growing trend towards student-centred collaborative learning and the development of appropriate curriculum designs. (5) Home-Based Learning: The pandemic has led to a rise in home-based learning, with students conducting experiments in their homes and learning at their own pace. (6) Teachers' Training: There is an increased focus on enhancing teachers' background, training, professional competencies, and interdisciplinary learning.

#### 4.1.2. Organisational Transformations in Higher Education in the Future

Transformations expected to continue over the next 5 to 10 years as higher education institutions adapt to the new normal post-COVID-19 are: (1) Digital Transformation: Higher education institutions are experiencing profound transformations, driven by the necessity to digitise education and training processes. This encompasses the adoption of novel technologies and pedagogical methods for online learning. (2) Resilience and Change Management: Institutions are formulating strategies to cope with unforeseen uncertainties during and post-pandemic. This involves building resilience and effectively managing change. (3) Curriculum Change: The pandemic has allowed educators and policymakers to reassess education systems and reimagine what is necessary for future generations. This could lead to substantial changes in curriculum design. (4) Sustainability: Higher education institutions are concentrating on sustainability, both in terms of their operations and the content of their curricula. (5) Increased Use of Online Learning: Once an auxiliary activity to the learning process, online education has become mainstream.

### 4.2. Second Round

A second questionnaire was formulated based on the responses from the first round. This round aimed to measure the extent of agreement among the respondents on the first

round's findings. Respondents could contribute additional information if they believed something was missing. The questionnaire comprised statements that the respondents rated on a five-point Likert scale, indicating the importance of the statements in their credit process. A score of five signified high importance, while a score of one indicated low importance. A key objective of the second round was to ascertain the level of consensus among the respondents' answers.

The results of the second round, as depicted in Table 1, showed high consensus (SD $\leq$ 1).

**Table 1.** Results of the second round of the Delphi study.

| Measurement | Average | SD | Consensus Level |
|---|---|---|---|
| **Question 1** | | | |
| Active Learning | 4.2 | 0.74 | High |
| Hybrid Learning | 4.3 | 0.65 | High |
| Digital Technologies | 4.8 | 0.63 | High |
| Student-Centred Collaborative Learning | 3.9 | 1.1 | High |
| Teachers' Training | 3.8 | 1.05 | High |
| **Question 2** | | | |
| Digital Transformation | 4.7 | 0.7 | High |
| Resilience and Change Management | 4.2 | 0.62 | High |
| Curriculum Change | 3.8 | 0.91 | High |
| Sustainability | 3.7 | 1.08 | High |
| Increased Use of Online Learning | 4.8 | 0.65 | High |

## 5. Discussion

The COVID-19 pandemic has reshaped the professional landscape, expediting the transition towards remote work and digital collaboration [65]. This paradigm shift has necessitated a re-evaluation of conventional work models and collaboration tools. Through its flexibility, remote work has broadened and diversified talent pools, enabling employers to recruit the most suitable candidates irrespective of geographical location. This has the potential to accelerate diversity within organisations [66].

However, it is crucial to acknowledge that remote work can enhance workplace diversity and also present unique challenges. Marginalised or underrepresented employee groups often need help with visibility and recognition for their contributions. The remote work environment can exacerbate these issues, potentially increasing exclusion [67]. Other challenges include the difficulty of disconnecting from work, leading to extended work hours and potential burnout, feelings of isolation, communication and collaboration difficulties, especially across different time zones, and increased cybersecurity risks. Furthermore, not all employees can access the necessary technology or stable internet connectivity to work effectively from home. Building and maintaining trust within remote teams can also be challenging.

The shift towards remote work has underscored the importance of digital skills, which are increasingly becoming a cross-sectoral requirement. The COVID-19 pandemic has amplified the demand for digital skills across various occupations, particularly non-ICT ones. Prioritising access to high-quality digital skills training for all workers and citizens has become a key policy focus. However, significant work remains in Continuing Vocational Education and Training (CVET) to bridge the digital skills gap among adults. Additionally, equipping teachers and trainers with digital competencies to effectively support learners is an area that requires further development in many national skill systems. It is also worth noting that the pandemic has disproportionately impacted individuals with low skills, and women have been more affected than men. Therefore, addressing these challenges

and ensuring equitable access to remote work opportunities is paramount. While remote work broadened talent pools and enhanced diversity, it posed challenges like visibility issues and increased cybersecurity risks. Additionally, the demand for digital skills surged, emphasising the need for high-quality training.

Our research findings underscore the significance of pedagogical and organisational characteristics such as adaptability, resilience, and a willingness to embrace digital transformation in preserving the long-term changes induced by pandemics. Our study identified several key trends shaping higher education in the post-COVID-19 era, including a shift towards active learning, the emergence of hybrid learning, the advancement of digital technologies, a growing trend towards student-centred collaborative learning, and a rise in home-based learning. These trends suggest rethinking traditional concepts of time and space in education, with a blend of virtual and physical classrooms emerging for immersive and experiential learning. Interestingly, these findings echo the working hypotheses of our study, affirming that the pandemic has indeed catalysed a shift towards more inclusive and diverse educational practices.

Our research findings underscore the significance of pedagogical and organisational characteristics such as adaptability, resilience, and a willingness to embrace digital transformation in preserving the long-term changes induced by pandemics. These results align with previous studies that emphasise the importance of these traits in navigating pandemic-induced challenges. Educators can practically apply theories like systems thinking by designing courses that consider interconnected content, using regular feedback loops to refine teaching strategies, and encouraging students to see the bigger picture. Additionally, integrating action research involves reflective practice, collaboration with colleagues, and involving students as co-researchers. Lastly, promoting digital literacy and creating inclusive online communities enhance the effectiveness of remote teaching. Adopting agile methodologies in higher education can be crucial for adapting to the new normal in a post-COVID education landscape. By emphasising flexibility, collaboration, and responsiveness, educators can better meet the evolving needs of students and navigate any future disruptions. Agile approaches empower institutions to create dynamic, student-focused learning environments that promote innovation and resilience in uncertainty. This shift towards agile education can help prepare students for success in a rapidly changing world and ensure the continuity of quality teaching and learning experiences [56]. Ricardo Abad Barros-Castro et al.'s theory on Computer-Supported Collaborative Learning (CSCL) within higher education has become increasingly relevant. By leveraging technology to enhance collaborative learning experiences, educators can facilitate meaningful interactions and knowledge construction among students in a virtual or hybrid learning environment. Barros-Castro et al.'s framework offers a systematic approach to evaluate and improve CSCL initiatives, providing practical insights that can help institutions adapt and enhance collaborative learning practices in the new educational paradigm shaped by the pandemic [57]. In light of our findings, educators are increasingly seeking deep learning approaches due to the effectiveness of active learning, student-centred collaborative learning, and the alignment of hybrid learning with deep learning principles. The rise in home-based learning necessitates adaptable, deep learning strategies that leverage digital tools for personalised, immersive experiences. These dimensions of deep learning are being validated and updated as education adapts to the changing needs of students in a digital world.

Contemporary higher education faces challenges brought about by the pandemic. Key trends include active learning, hybrid models, digital technologies, and student-centred collaborative approaches. Simultaneously, we must rethink traditional notions of time and space in education. Systems thinking and action research are tools to address these challenges and shape a more inclusive and diverse educational environment.

## 6. Conclusions

The research focused on exploring current trends and anticipated changes in higher education, specifically within the context of the COVID-19 pandemic. While numerous studies have explored the impact of the pandemic on education, most have relied on data gathered through questionnaires or sourced from educators and students. This approach could overlook several crucial aspects of the transformations occurring within higher education institutions. This knowledge gap necessitated a comprehensive understanding of the changes and trends observed by experts within these institutions.

A Delphi study was conducted with 12 professors to identify the key trends and transformations in higher education in the post-COVID-19 era. Critical trends for each of the two questions were identified as significant in shaping the future of higher education. Question 1's trends included Active Learning, Hybrid Learning, Digital Technologies, Student-Centred Collaborative Learning, and Teachers' Training. For Question 2, the transformations included Digital Transformation, Resilience and Change Management, Curriculum Change, Sustainability, and Increased Use of Online Learning. As anticipated, the most significant trends and transformations were those related to adopting digital technologies and the shift towards more flexible and student-centred learning approaches.

The professors also identified several additional factors that should be considered when examining the future of higher education. These included pedagogical and organisational characteristics such as the ability to adapt to changes, the capacity for resilience, and the willingness to embrace digital transformation.

The study's limitations include the potential lack of generalisability due to a small, specific sample, meaning that the findings may not apply to all higher education contexts. Additionally, the results may not accurately reflect the experiences and challenges of professors in different countries or cultural settings. The research failed to consider the benefits of incorporating artificial intelligence (AI) into education, which could improve personalised learning, offer round-the-clock support, and facilitate data-driven decision-making. Addressing these factors in future studies could enhance the comprehension of evolving trends in higher education post-pandemic. Future research should focus on exploring the impact of integrating AI in education, understanding the perspectives of agile educators, and analysing the characteristics of agile classrooms to further enrich our understanding of post-pandemic higher education trends. Additionally, future research should delve deeper into these trends and transformations, especially within diverse geographical and institutional contexts. We recommend expanding the research beyond Slovenia by comparing the findings with professors from diverse countries or regions. Consideration of cultural disparities, institutional settings, and policy differences is crucial. Additionally, exploring whether the trends observed in Slovenia align with global standards or if there are distinctive elements unique to the Slovenian higher education system is essential. Furthermore, conducting interviews or surveys with students could provide valuable insights into their remote learning experiences and perspectives. Such exploration will contribute to a more holistic and nuanced comprehension of higher education's trajectory in the post-COVID-19 era.

**Author Contributions:** F.Q. conceptualisation, writing—review and editing; S.K. conceptualisation, writing—review and editing; M.P.B. supervision, methodology; M.M. writing—original draft preparation, methodology. All authors have read and agreed to the published version of the manuscript.

**Funding:** The University of Maribor, Faculty of Organizational Sciences and the authors acknowledge the financial support from the Slovenian Research and Innovation Agency (research core funding P5-0018).

**Data Availability Statement:** The data will be available from the corresponding author upon request.

**Conflicts of Interest:** The authors declare no conflicts of interest.

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
