# Peer review of "Slovenian Higher Education in a Post-Pandemic World: Trends and Transformations"

_systems, doi:10.3390/systems12040132_

Round 1
Reviewer 1 Report
Comments and Suggestions for Authors
Thank you for allowing me to review the manuscript entitled “Higher Education in a Post-Pandemic World: Trends and Transformations.” The topic regarding the changes within the higher education field during and after the pandemic is very important. Here are some comments.
Title: maybe authors should consider adding the title "Slovenian higher education ...", as the study was conducted with professors from Slovenia.
Abstract: gives all the necessary information, about the purpose, data, method, results, and implications briefly.
Introduction: In a comprehensive introduction, the authors present the problem under consideration, outline the background, and justify the need for such research.
Literature review: The literature review offers us a detailed overview of current and relevant sources that have addressed various aspects related to the transformation of higher education due to Covid19.
Methodology: The sample is consistent with this type of studies. Perhaps authors could describe the method of selecting and contacting the selected persons. The implementation procedure of the Delphi method can be placed in time and describe the way of communication.
Results, discussion, and conclusion: no comments for these parts of the article. The authors supported the results with secondary literature and provided directions for future research. They conclude that the most significant trends and transformations are related to adopting digital technologies and the shift towards more flexible and student-centered learning approaches.
Author Response
Dear reviewer,
I sincerely appreciate your time and effort in reviewing our work. Thank you for your valuable feedback and the corrections you provided. We have carefully considered your suggestions and made the necessary adjustments.
Slovenian Higher Education in a Post-Pandemic World: Trends and Transformations We add in the title "Slovenian ...", as the study was conducted with professors from Slovenia. We describe the method of selecting and contacting the selected persons. We mention inclusion/exclusion criteria, age, gender, or field of study of the participants. Another aspect I would like to suggest the authors consider is providing more information about the questionnaire. In this study, the sample consisted of twelve professors from Slovenia who are experts in higher education. The participants included 5 women and 7 men, all from the fields of social sciences. They were selected based on their extensive research, professional experiences, and significant contributions to the field. We selected professors with a minimum academic rank of associate professor and at least 10 years of teaching experience in higher education institutions to ensure they have a deep understanding of the subject matter. Additionally, we considered professors who have actively participated in online teaching or remote work during the COVID-19 pandemic to gather firsthand experiences and insights on the challenges and successes of this transition. Their diverse perspectives and insights provide a rich understanding of the future trends and possibilities of remote work and digital collaboration in higher education post-COVID-19. Best regards, authors
Reviewer 2 Report
Comments and Suggestions for Authors
Dear authors,
Thank you for the opportunity to read your article. I must emphasize that I like your research perspective and the steps applied. Hence I also have some concerns regarding the utility of the article. In order to improve the article you may at least add an exploratory factor analysis on the items discovered and collected in table 1.
The weakest point of the article is the representativeness of the sample and the fact that the authors do not take into account the agile educator perspective and agile classroom characteristics... This is a rather current trend that will evolve and the authors didn't mention it at all.....!!!!! They also didn't take into account the advantages of using AI in education (https://facilitate-ai.eu/). Please ask different teachers for their opinions and reshape the statistical analysis.
Thus you can come up with great solutions for future education and potentiate the value of the article, in order to obtain a sustainable education.
Success!
Author Response
Dear reviewer,
Thank you for your thoughtful feedback on our manuscript. We appreciate your positive remarks regarding our research perspective and methodology. We have taken your suggestions seriously and made several important revisions to enhance the utility of the article.
We have made significant changes to the manuscript, addressing the sampling issues and incorporating an agile perspective. Additionally, we have set clear limitations, and the track changes are attached for your reference.
Thank you for your valuable insights.
Best regards,
authors
Reviewer 3 Report
Comments and Suggestions for Authors
The text can be seen as one with a novel approach, as it looks forward rather than backward concerning the pandemic situation and its impact on education. It forecasts what will or can happen based on the experiences teachers have. The theoretical background is extensive and relatively broad. However, more detailed information on the sample is missing. The authors used vague descriptions, stating, "twelve professors from Slovenia who are experts in higher education. These individuals were carefully selected based on their extensive research, professional experiences, and significant contributions to the field." The authors do not mention inclusion/exclusion criteria, age, gender, or field of study of the participants. Another aspect I would like to suggest the authors consider is providing more information about the questionnaire. Again, the information provided is very brief, with no details on the number of questions/statements, perhaps the area covered, or the statements themselves.
Author Response
Dear reviewer,
I sincerely appreciate your time and effort in reviewing our work. Thank you for your valuable feedback and the corrections you provided. We have carefully considered your suggestions and made the necessary adjustments.
We describe the method of selecting and contacting the selected persons.We mention inclusion/exclusion criteria, age, gender, or field of study of the participants. Another aspect I would like to suggest the authors consider is providing more information about the questionnaire. In this study, the sample consisted of twelve professors from Slovenia who are experts in higher education. The participants included 5 women and 7 men, all from the fields of social sciences. They were selected based on their extensive research, professional experiences, and significant contributions to the field. We selected professors with a minimum academic rank of associate professor and at least 10 years of teaching experience in higher education institutions to ensure they have a deep understanding of the subject matter. Additionally, we considered professors who have actively participated in online teaching or remote work during the COVID-19 pandemic to gather firsthand experiences and insights on the challenges and successes of this transition. Their diverse perspectives and insights provide a rich understanding of the future trends and possibilities of remote work and digital collaboration in higher education post-COVID-19. Questionnaire: In the second stage of the study, a structured questionnaire was developed based on the responses from the initial stage. The questionnaire included specific questions related to the key themes and trends identified during the open-ended discussions. Participants were asked to rate the importance of various factors influencing the future of remote work and digital collaboration in higher education, as well as to provide their views on the challenges and opportunities presented by these trends. The data collected from both stages of the study was analyzed using qualitative and quantitative methods to identify consensus views and emerging patterns. The findings from the study were used to inform future strategies and decision-making in higher education institutions in response to the evolving landscape of remote work and digital collaboration post-COVID-19.
Best regards, authors
Round 2
Reviewer 2 Report
Comments and Suggestions for Authors
Dear authors the antiplagiarism report is 19%.
In my opinion is too high!
Author Response
Dear Reviewer,
thank you for your feedback on our article.
In response to your concern about the plagiarism report, we have diligently revised the content to reduce similarities. The updated report now reflects a 9% similarity.
Additionally, we have attached a screenshot of the revised plagiarism report for your reference.
Best regards,
authors

Reviewer 3 Report
Comments and Suggestions for Authors
The authors added necessary information and the academic quality of text has been improved.
Comments on the Quality of English LanguageThe academic quality of the text has been improved, and the language is now understandable and at an acceptable level. Both the lexical and syntactic levels are well-managed.
Author Response
Dear Reviewer,
thank you for your insightful feedback on our article. We appreciate your thorough evaluation.
In response to your comments, we have done minor editing of English (track changes in the final text).
Best regards,
authors